# An evolutionary functional genomics approach identifies novel candidate regions involved in isoniazid resistance in *Mycobacterium tuberculosis*

Victoria Furió [1,4 ✉], Miguel Moreno-Molina [1,4], Álvaro Chiner-Oms [1], Luis M. Villamayor[2], Manuela Torres-Puente[1] & Iñaki Comas [1,3]

Efforts to eradicate tuberculosis are hampered by the rise and spread of antibiotic resistance. Several large-scale projects have aimed to specifically link clinical mutations to resistance phenotypes, but they were limited in both their explanatory and predictive powers. Here, we combine functional genomics and phylogenetic associations using clinical strain genomes to decipher the architecture of isoniazid resistance and search for new resistance determinants. This approach has allowed us to confirm the main target route of the antibiotic, determine the clinical relevance of redox metabolism as an isoniazid resistance mechanism and identify novel candidate genes harboring resistance mutations in strains with previously unexplained isoniazid resistance. This approach can be useful for characterizing how the tuberculosis bacilli acquire resistance to new antibiotics and how to forestall them.

[1] Institute of Biomedicine of Valencia (IBV-CSIC), Valencia 46020, Spain. [2] FISABIO Public Health (CSISP), Valencia 46010, Spain. [3] CIBER in Epidemiology and Public Health, Madrid 28029, Spain. [4] These authors contributed equally: Victoria Furió, Miguel Moreno-Molina. ✉email: vfurio@ibv.csic.es

In 2018, an estimated 484,000 people contracted drug-resistant tuberculosis and a further 214,000 people died from it[1]. Resistance to antitubercular drugs has been present ever since their introduction decades ago but it is now becoming a pressing problem, as it hampers our ability to control and eradicate the disease. Drug-resistant tuberculosis requires longer treatments, has lower cure rates, and spreads in the population, particularly in high-burden countries[1]. Licensing new antibiotics is not a definitive solution as the bacteria can develop resistance to those antibiotics as well[2,3]. A new approach is needed in which a thorough understanding of the evolutionary forces shaping resistance helps us understand how it is acquired and how it can be reversed.

Most of what we know of tuberculosis drug resistance comes from genetic association studies in which a particular mutation is associated with a specific resistance phenotype[4]. We now have large databases of diagnostic mutations with which we can reliably predict the resistance phenotype of our strain when we determine its genomic sequence[5]. For instance, we can detect rifampicin resistance with a 92% sensitivity, but the figure drops to 87% for isoniazid and 58% for ethambutol[6]. However, there is still a knowledge gap as the catalog of mutations is incomplete and we do not know most of the resistance-causing mutations and mechanisms for some antibiotics. To close this gap, there are a series of ongoing efforts by consortiums like ReSeqTB and CRyPTIC, wherein tens of thousands of isolates are being phenotyped and genotyped in order to obtain a comprehensive mutation database with the overarching aim to develop new diagnostic assays with maximum specificity and sensitivity. However, we still need more than mutation databases to effectively combat drug resistance. First, it is impossible to predict the phenotype for a mutation never seen before. For this reason, it is very difficult to accurately predict resistance to newly licensed antibiotics. In addition, an approach that prioritizes diagnostic mutations generally provides very little information on other mutations that contribute to the resistant phenotype but are normally overlooked, because their clinical effect is small or they are in genes not known to be associated with resistance. Finally, we need extensive insight on the genetic architecture of resistance and especially on any changes that can increase sensitivity to the antibiotic. This is important, as this information could be used to find companion drugs that potentiate the action of antibiotics or that prevent or even reverse resistance[7].

One way to unveil the genetic basis of resistance is by means of functional genomics, such as transposon mutagenesis approaches. This technique involves the genetic alteration of every gene in the genome for explicit genotype–phenotype associations[8,9], thus revealing more genetic determinants than regular association studies do. This approach successfully overcomes the shortcomings of genetic association studies: it can be used in a prospective way, as it involves the systematic generation and testing of resistant mutants; it can detect both genes with large and small effects on resistance; and it explicitly detects genes that increase sensitivity when disrupted, thus indicating which genes are most promising for treatments to prevent or reverse the evolution of resistance. However, transposon mutagenesis alters the gene by disrupting it, highly informative about the biology of resistance but limited in clinical explanation potential, as most type of mutations found in clinical resistance of *Mycobacterium tuberculosis* are single-nucleotide polymorphisms (SNPs). Conversely, the low diversity of the *M. tuberculosis* Complex (MTBC), its clonality, and the fact that clinical resistance is encoded in the chromosome makes *M. tuberculosis* amenable for phylogenetic association tests[10], which can determine which mutations are associated with resistance in the bacterial phylogeny and are thus clinically relevant.

In this study, we provide a combined approach that uses functional genomics and phylogenetic inference from clinical data to provide an in-depth picture of resistance to the first-line antibiotic isoniazid. Isoniazid is a well-studied drug, yet we are still unable to determine the causal mutation in around 6% of resistant strains[11],

although some researchers have reported up to 25% in certain settings[12]. Here we systematically determine the effect on isoniazid resistance of every non-essential gene in the tuberculosis genome using transposon sequencing (TnSeq) and afterwards we use clinical data to find out which of those genes are more likely to harbor resistance mutations. We successfully find novel regions associated with increased resistance in vitro, determine two major resistance pathways for the mode of action of the antibiotic and identify novel associated regions to clinical resistance not described before. We believe this approach will help uncover the resistance determinants for poorly studied antibiotics, as well as deepen our understanding of resistance emergence, spread, and evolution.

## Results

**Functional genomics allows for detection of resistance-associated genomic regions**. We generated a highly saturated *M. tuberculosis* H37Rv pool with over 100,000 different transposon-insertion mutants following the protocol by Long et al.[8]. Using TnSeq, we found 58,389 out of 74,603 possible insertion sites had at least one read, meaning a saturation of 78%. For comparison, a systematic study with 14 independent pools found saturations in the range of 42% to 64% and a combined saturation of 84.3%[13], meaning that our pool is highly saturated. In addition, as many as 43% of our non-inserted sites and only 0.03% of our inserted sites were in regions described as essential in that study. The pool also showed a tenfold increase in the frequency of bacteria resistant to isoniazid compared to the original clone (Supplementary Fig. 1a).

The pool was tested in duplicate with a subinhibitory dose of isoniazid close to the IC50 for 13 generations (Fig. 1a). We expected this specific dose of isoniazid to provide intermediate levels of selection and to maximize the number of genomic features detected. Optical density measurements showed that isoniazid was partially inhibiting bacterial growth (Fig. 1b). In the presence of isoniazid, the proportion of isoniazid-resistant bacteria increased 100-fold to 1000-fold, whereas control cultures showed no change (Supplementary Fig. 1b).

We determined the frequencies of the different insertion mutants in all four experimental populations using TnSeq (Supplementary Data 1). Isoniazid-treated populations had a higher proportion of sites with null frequency and the top 100 sites comprised a larger share of the total counts (Fig. 1c). We transformed the normalized data into standardized fitness measurements, which can be directly compared between populations. We defined resistance as the net change in fitness in the presence of the antibiotic and calculated it as the difference between fitness in the presence and absence of the antibiotic for each insertion site (Supplementary Data 1). Insertion mutants for *katG*, the gene most frequently involved in isoniazid resistance, were disproportionately overrepresented in antibiotic-treated populations and thus displayed very high resistance values. All these results show that the selection step had the intended effect.

It is important to note that transposon libraries have limitations, as they only allow us to study the effect of gene disruptions. This has two main consequences: (i) we cannot study essential genes, as they cannot tolerate insertion, and (ii) we cannot observe the effect of more subtle genetic changes such as single-nucleotide mutations. To overcome these limitations, we used two main approaches: first, we used functional and pathway analysis to understand which portions of bacterial metabolism were involved in isoniazid resistance and, second, we used phylogenetic association to determine which genes were accumulating mutations in clinical settings.

We analyzed all insertion sites with an annotation-aware sliding window approach to find changes in resistance that were consistent over stretches of the genome independent of the size of

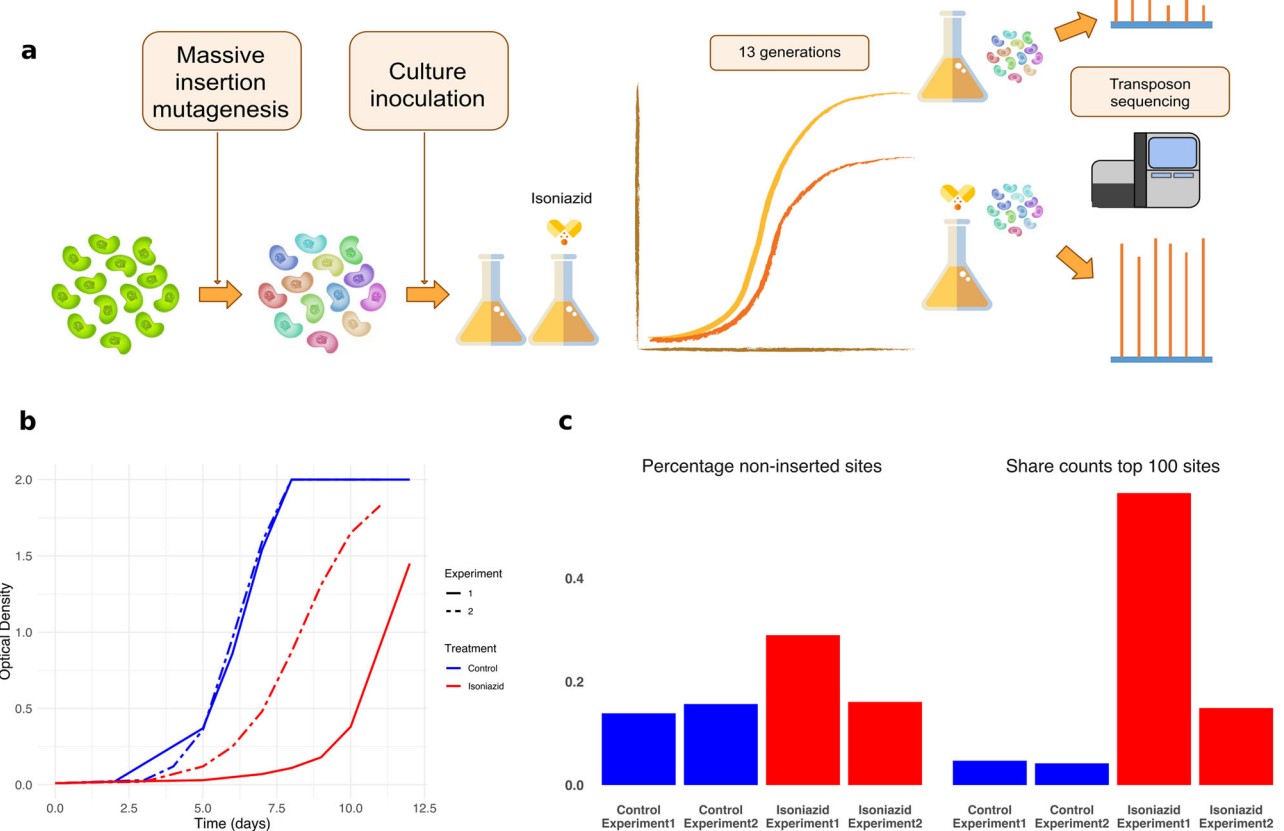

**Fig. 1 Transposon sequencing detects changes in response to isoniazid. a** Design of the experiment. Parallel antibiotic-containing and antibiotic-free cultures were inoculated with a saturated insertion mutant pool. After ~13 generations, bacterial DNA was extracted and sequenced to determine the relative abundance of each mutant. **b** Optical density of the different cultures throughout the experiment. Graph shows that isoniazid partially inhibits bacterial growth. **c** Isoniazid-containing cultures show strong enrichment of a fraction of insertions indicating a selective advantage relative to the bulk of the population, showing that those cultures experienced higher levels of selection (blue = control, red = isoniazid experiment).

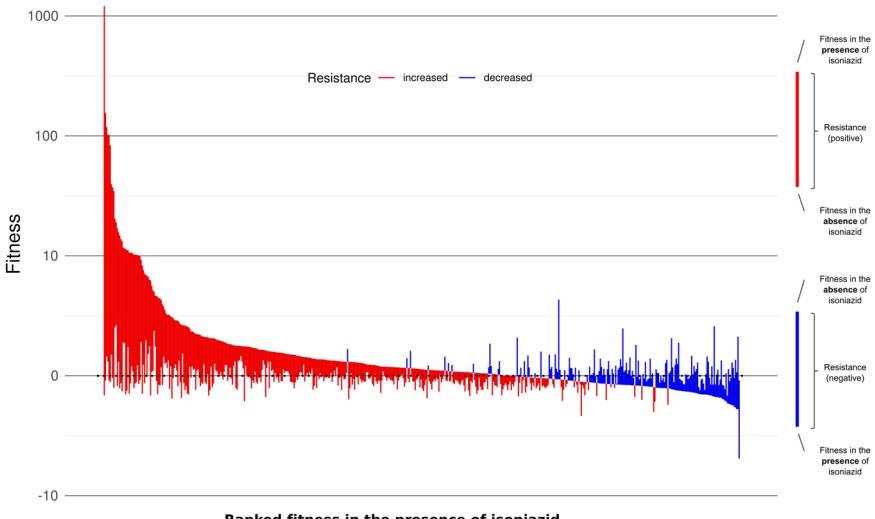

**Fig. 2 Isoniazid resistance is associated with multiple genomic regions.** Median fitness in the presence and absence of isoniazid and median resistance for all resistance-altering genes, ordered in decreasing fitness in the presence of the antibiotic (red = increased resistance, blue = decreased resistance). Most genes that increased resistance when disrupted also had a higher fitness in the presence of the antibiotic.

the effect. We detected a total of 555 genes and intergenic regions that alter isoniazid sensitivity when disrupted (resistance-altering genomic features, Supplementary Data 1). Of those regions, 411 were associated with increased resistance, whereas 144 were

associated with increased sensitivity (resistance-increasing and sensitivity-increasing features, respectively). Figure 2 depicts these regions ordered by their fitness in the presence of the antibiotic. Given that fitness in the presence of the antibiotic is the primary

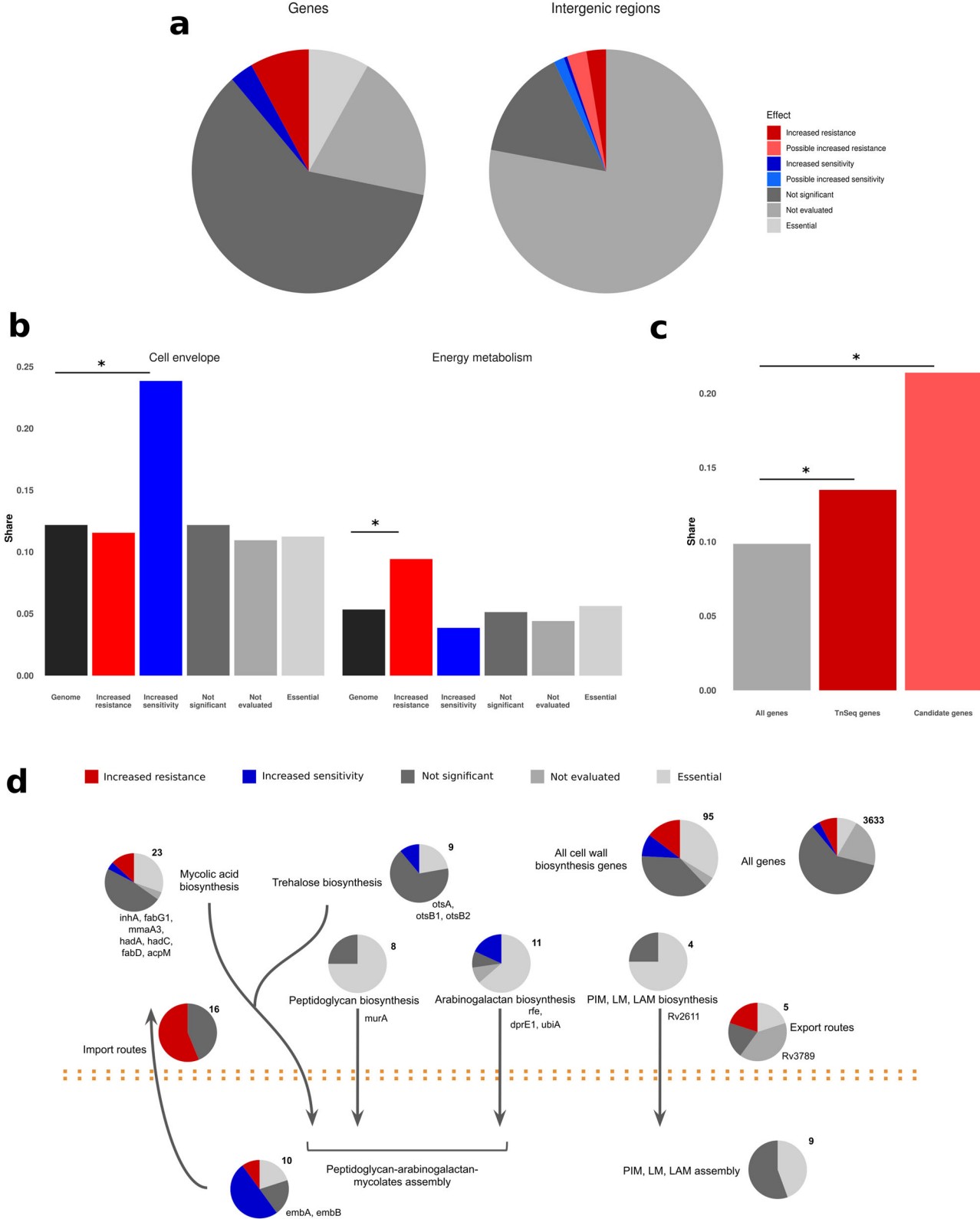

**Fig. 3 Genes associated with isoniazid resistance follow definite functional patterns. a** Classification of all genes and intergenic regions in the *M. tuberculosis* genome according to the effect of insertions on resistance. Essential genes were obtained from DeJesus et al.[13]. **b** Genes associated with increased sensitivity were enriched in cell envelope genes (*n* = 130), whereas those associated with increased resistance were enriched in energy metabolism genes (*n* = 329). **c** Oxidoreductases were even more enriched in the final candidates than they were in the genes detected with functional genomics (*n* = 57 and 459). **d** Cell wall biosynthesis genes were enriched in features associated to resistance both functionally and phylogenetically to resistance.

driver of the resistance phenotype, we observed resistance was split into two groups according to whether the genes conferred increased sensitivity or resistance when disrupted. Multiple features showed a significant change in resistance, implying that they could, in theory, confer clinically relevant resistance in vivo when mutated. Features that could be tested but showed no significant effect were considered non-associated features. Our method did not allow us to test regions that had fewer than six inserted TA sites (genomic regions where the sequence is exactly 'TA'), although not all of these regions were essential. Intergenic regions tend to be small and often harbor regulatory sequences for the genes they precede but were massively overrepresented in non-evaluated features. Thus, we can assume that intergenic regions preceding candidate genes and with resistance scores that show the same sign as those in the gene are probably associated with resistance. Using this approach, we found 126 additional probable resistance-altering features, 82 of which were associated with increased resistance (Fig. 3a). Among resistance-altering features we found several regions known to be associated with clinical isoniazid resistance, such as *katG*, *ahpC* and its promoter region, and *fabG1* and its promoter region[14]. Our results were also consistent with similar data from Xu et al.[15], further confirming that resistance-altering features are associated with isoniazid resistance (see "Methods").

**Genes associated with altered resistance follow definite functional patterns.** We noticed that resistance-altering features tended to group together on the genome. One explanation for this observation is that functionally related genes sometimes cluster in operons, so they can be transcribed together. To test this we obtained the H37Rv operon annotations from BioCyc[16] and used a sampling approach, finding that significant genes clustered in operons more than expected by chance in 100,000 random samples (388 transcription clusters vs. at least 395 in the simulations, $p < 10^{-5}$). In partiular, two large operons nearly entirely comprised resistance-altering features (Fisher's exact test, $p < 0.01$): *nuo* and *mce1*. This proves that these features are not randomly distributed around the genome but show at least some functional relatedness to one another.

We further hypothesized that resistance depends on specific cellular processes. To test this at the most general level, we compared the relative shares of both versions of the TubercuList functional categories[17,18] in the resistance-altering features with their global shares (Fig. 3b and Supplementary Data 2). We found that although all categories were represented, resistance-increasing features were enriched in energy metabolism (Fisher's exact test, $p = 0.0029$), intermediary metabolism and respiration (Fisher's exact test, $p = 0.012$), and virulence genes (Fisher's exact test, $p = 0.017$), whereas sensitivity-increasing features were enriched in cell envelope (Fisher's exact test, $p = 0.0002$), and cell wall and cell process genes (Fisher's exact test, $p = 0.00001$).

To further understand the genetic architecture of isoniazid resistance, we conducted a pathway enrichment analysis using data from both Kyoto Encyclopedia of Genes and Genomes[19] and BioCyc using resampling (Supplementary Data 2). Results were consistent using both databases and revealed that pathways associated with the mycolic acids and cell wall biosynthesis were significantly enriched among sensitivity-increasing features (1000 samples, $p < 0.05$). This result is not surprising, given that isoniazid interferes with the biosynthesis of mycolic acids, one of the main components of the bacterial envelope, although the main mycolate biosynthesis pathway itself was not significantly enriched ($p = 0.09$). Finally, we collated a data set with all cell wall biosynthesis genes (Supplementary Data 2) from published sources[20] and confirmed that sensitivity-increasing features were

enriched in those genes (Fisher's exact test, $p = 0.0001$). Resistance-increasing features were enriched as well (Fisher's exact test, $p = 0.0069$), but that depended mainly on the *mce1* operon as shown in Fig. 3d. In contrast, sensitivity-increasing genes can be found all over the cell wall biosynthesis pathway, which demonstrates the central role of the cell envelope in intrinsic resistance and in isoniazid resistance in particular.

Among resistance-increasing genes, oxidative metabolism was enriched as well ($p < 0.001$) with the electron transport chain as the most enriched pathway overall ($p < 0.001$). In addition, nicotinate and nicotinamide metabolism was enriched for both resistance-increasing and sensitivity-increasing genes ($p < 0.02$), suggesting that NADH metabolism might be of importance. To ensure that the enrichment in oxidative metabolism pathways did not depend entirely on the *nuo* operon, we mined the genomic annotation for H37Rv from the National Center for Biotechnology Informatio (NCBI) using the terms: oxidoreductase, oxidase, reductase, redox, peroxidase, dehydrogenase, NAD, NADH, NADP, and NADPH, and we flagged any feature that contained any of those terms in their name or function description as redox-associated. We found that resistance-increasing genes were significantly enriched in oxidative metabolism genes (Fisher's exact test, $p = 0.03$; Fig. 3c). All these observations point to redox metabolism having a role in isoniazid resistance.

**A phylogenetic association test identifies candidate regions associated with clinical resistance.** So far, we have successfully linked resistance-altering features to isoniazid resistance at an in vitro and functional level, but we still do not know what their importance in a clinical setting is. We used a phylogenetic test to identify regions associated with resistance in clinical strains. We first set out to obtain a phylogeny that encompassed tuberculosis strain variability using 4762 globally distributed, published *M. tuberculosis* complex genomes (Supplementary Data 3) with around 240,000 polymorphic sites (Supplementary Fig. 2). The data set included 32% of strains resistant to at least one drug. We reconstructed the evolutionary history for each variable site inferring how many substitution events had occurred and where in the phylogeny they had taken place (Supplementary Data 4). Finally, we sought to determine which regions in the whole genome are more strongly associated with resistance by calculating the PhyC parameter[21], which acts as an association test and measures the degree of mutation accumulation for a particular gene in predetermined branches of the phylogeny. We first determined in which specific branches an antibiotic-resistance mutation had occurred using a comprehensive list of resistance mutations based on PhyResSE[22] and ReSeqTB (Supplementary Data 5). We then tested which mutations tend to appear in resistant versus susceptible subtrees by random sampling.

We identified 511 regions significantly associated with antibiotic resistance in the phylogeny ($p < 0.05$, Supplementary Data 4). Most of the top scoring regions were already known resistance genes for first- and second-line antibiotics (Supplementary Data 4), which shows that there are still many unidentified resistance mutations in those genes. Some other top scoring regions are known to be associated with compensatory mutations, which were also expected to appear after resistance mutations to compensate for their cost. Thus, phylogenetic association does a good job in identifying genes known to be relevant to antibiotic resistance.

**Novel isoniazid resistance determinants were identified by combining functional genomics and phylogenetic association.** We combined our functional data on isoniazid resistance with phylogenetic convergence results to look for isoniazid resistance

**Table 1 Candidate genes.**

| Rv number | Call | Name or associated gene | Function |
|---|---|---|---|
| Rv0001 | IR | dnaA | Chromosomal replication initiator protein, regulates chromosomal replication |
| Rv0010c | IS | Rv0010c | Conserved membrane protein |
| IG_Rv0020c_Rv0021c | IR | fhaA | Conserved hypothetical protein, thought to be involved in signal transduction |
| Rv0134 | IR | ephF | Epoxide hydrolase, thought to be involved in detoxification reactions following oxidative damage to lipids |
| IG_Rv0237_Rv0238 | IS* | Rv0238 | Transcriptional regulator, tetR-family |
| Rv0392c | IR | ndhA | Membrane NADH dehydrogenase, transfer of electrons from NADH to the respiratory chain |
| Rv0450c | IR | mmpL4 | Transmembrane transport protein, thought to be involved in fatty acid transport |
| Rv0740 | IR | Rv0740 | Conserved hypothetical protein |
| IG_Rv0767c_Rv0768 | IR | aldA | Aldehyde dehydrogenase NAD-dependent |
| Rv0994 | IR | moeA1 | Molybdopterin biosynthesis protein |
| Rv1022 | IS | lpqU | Lipoprotein |
| Rv1053c | IR | Rv1053c | Hypothetical protein |
| Rv1086 | IS | Rv1086 | Short-chain Z-isoprenyl diphosphate synthase, catalyzes the first committed step in the synthesis of decaprenyl diphosphate, a molecule that has a central role in the biosynthesis of most features of the mycobacterial cell wall |
| Rv1194c | IR | Rv1194c | Conserved hypothetical protein |
| IG_Rv1364c_Rv1365c | IR* | Rv1364c | Conserved hypothetical protein |
| IG_Rv1482c_Rv1483 | IS* | fabG1 | 3-Oxoacyl-[acyl-carrier protein] reductase, involved in the fatty acid biosynthesis pathway (first reduction step, mycolic acid biosynthesis). Secondary isoniazid resistance gene |
| Rv1504c | IR | Rv1504c | Conserved hypothetical protein |
| Rv1512 | IR | epiA | Nucleotide-sugar epimerase |
| Rv1692 | IR | Rv1692 | Phosphatase |
| Rv1767 | IR | Rv1767 | Conserved hypothetical protein |
| IG_Rv1773c_Rv1774 | IR* | Rv1773c | Transcriptional regulator |
| Rv1780 | IR | Rv1780 | Conserved hypothetical protein |
| Rv1830 | IR | Rv1830 | Conserved hypothetical protein |
| Rv1836c | IS | Rv1836c | Conserved hypothetical protein |
| IG_Rv1843c_Rv1844c | IR* | guaB1 | Inosine-5-monophosphate dehydrogenase |
| IG_Rv1900c_Rv1901 | IS* | cinA | Competence damage-inducible protein A |
| Rv1905c | IR | aao | D-Amino acid oxidase |
| Rv1908c | IR | katG | Catalase-peroxidase-peroxynitritase T, main isoniazid resistance gene |
| Rv1928c | IR | Rv1928c | Short-chain type dehydrogenase/reductase |
| Rv2021c | IR | Rv2021c | Transcriptional regulator |
| IG_Rv2208_Rv2209 | IR | Rv2209 | Conserved membrane protein |
| Rv2214c | IR | ephD | Short-chain type dehydrogenase, thought to be involved in detoxification reactions following oxidative damage to lipids |
| Rv2333c | IR | stp | Conserved membrane transport protein, involved in transport of drug across the membrane (export) |
| Rv2386c | IR | mbtI | Isochorismate synthase, involved in mycobactin siderophore construction |
| IG_Rv2427A_Rv2428 | IS* | ahpC | Alkyl hydroperoxide reductase C protein, involved in oxidative stress response and secondary isoniazid resistance gene |
| Rv2428 | IS | ahpC | Alkyl hydroperoxide reductase C protein, involved in oxidative stress response and secondary isoniazid resistance gene |
| IG_Rv2560_Rv2561 | IR | Rv2561 | Conserved hypothetical protein |
| IG_Rv2709_Rv2710 | IR* | sigB | RNA polymerase sigma factor |
| Rv2710 | IR | sigB | RNA polymerase sigma factor |
| Rv2886c | IR | Rv2886c | Resolvase |
| Rv2994 | IS | Rv2994 | Conserved membrane protein, could be involved in efflux system |
| Rv3154 | IR | nuoJ | NADH dehydrogenase I chain J |
| IG_Rv3210c_Rv3211 | IR | rhlE | ATP-dependent RNA helicase, has a helix-destabilizing activity |
| IG_Rv3213c_Rv3214 | IS* | gpm2 | Phosphoglycerate mutase |
| Rv3229c | IS | desA3 | Linoleoyl-CoA desaturase, thought to be involved in lipid metabolism |
| IG_Rv3260c_Rv3261 | IR | fbiA | F420 biosynthesis protein |
| Rv3268 | IS | Rv3268 | Conserved hypothetical protein |
| Rv3272 | IR | Rv3272 | Conserved hypothetical protein |
| Rv3278c | IS | Rv3278c | Conserved membrane protein |
| Rv3490 | IR | otsA | Alpha, alpha-trehalose-phosphate synthase, involved in osmoregulatory trehalose biosynthesis |
| Rv3501c | IR | yrbE4A | Hypothetical membrane protein |
| Rv3600c | IS | Rv3600c | Conserved hypothetical protein |
| Rv3777 | IR | Rv3777 | Oxidoreductase |
| Rv3788 | IR | Rv3788 | Hypothetical protein |

**Table 1 (continued)**

| Rv number | Call | Name or associated gene | Function |
|---|---|---|---|
| Rv3789 | IR | Rv3789 | Conserved membrane protein |
| Rv3843c | IR | Rv3843c | Conserved membrane protein |
| Rv3908 | IR | *mutT4* | Conserved hypothetical protein, possible mutator protein? |

For intergenic regions, the neighboring gene most probable to be regulated by the region is given.
*IR* increased resistance, *IR\** probable increased resistance, *IS* increased sensitivity, *IS\** probable increased sensitivity.

candidate genes. Our reasoning was that if resistance-altering regions from our TnSeq experiment accumulated changes specifically in association with resistance mutations then they would probably be involved in the evolution of isoniazid resistance. We found 57 resistance-altering features that had more mutations occuring in resistant subtrees than expected (Table 1). Four of them were well-known isoniazid resistance determinants or associated regions (*katG*, *ahpC* and its promoter region, and the promoter region of *fabG1*), which still showed association even though diagnostic mutations had already been removed, thus confirming that the catalog of mutations conferring isoniazid resistance in those features is far from complete. This finding is in agreement with the frequent identification of unidentified, but rare, mutations in *katG* associated with isoniazid resistance in different settings[12,23].

These candidate resistance features are functionally diverse, showing the different ways in which *M. tuberculosis* can adapt to antibiotics (Table 1). Looking the genome annotation, the probable mechanisms operating here include increased efflux/decreased influx of the antibiotic (*mmpL4*, *stp*), altered transcriptional regulation (*sigB*), mycolic acids biosynthesis (*fabG1*), changes in NADH balance (*ndhA*, *nuoJ* among others), and increased mutagenesis due to changes in DNA repair (*mutT4*), among others. Some of the candidate features have no known function, which means that our strategy allows for discovery of new resistance determinants even if they are poorly characterized.

Only one of the candidate regions (*fabG1*) was involved in cell wall biosynthesis pathways but a further 16 out of the 95 genes in cell wall biosynthesis pathways showed phylogenetic association with resistance, which is a significantly enriched fraction (Fisher's exact test, *p* < 0.016, Supplementary Data 4 and Fig. 3c). We do not have functional data for some of these regions as they are essential and cannot tolerate insertion, but mutations in these genes probably also affect isoniazid resistance as they are in the same pathway as the antibiotic target itself and our TnSeq data show that cell wall biosynthesis pathways are enriched in genes functionally associated with isoniazid resistance. These results highlight the importance of cell wall biosynthesis in isoniazid action and resistance, demonstrating that functional genomics is a powerful tool for discovering important pathways or even determining the mode of action.

We found that 8 out of 42 candidate genes were associated with redox metabolism. This result was mainly due to resistance-increasing genes, which accounted for seven of the eight redox genes and represented a significantly enriched fraction (Fisher's exact test, *p* = 0.024; Fig. 3c). In contrast, genes phylogenetically associated with resistance as a whole were not enriched in redox genes (44 out of 377 genes; Fisher's exact test, *p* = 0.09). In addition, 3 of the 15 candidate intergenic regions are next to the start of a redox gene. These results confirm that redox metabolism plays a clinically relevant role in the evolution of isoniazid resistance.

We confirmed that the resistance phenotype inferred from the TnSeq assay was associated with the expected change in sensitivity by determining the minimum inhibitory concentrations (MICs) for a representative sample of the candidate genes using the resazurin microdilution assay. For this experiment, we used Bacillus Calmette-Guérin (BCG) Danish insertion mutants from the BCCM/ITM

Mycobacteria Collection. Our results showed that mutants with insertions in resistance-increasing features had higher MICs than mutants with insertions in either sensitivity-increasing (Wilcoxon test, *p* = 0.0346) or nonsignificant (Wilcoxon test, *p* = 0.0202) features (Supplementary Fig. 4 and Supplementary Data 6). The result remained significant even when IC50s were used (Wilcoxon test, *p* = 0.0036 and *p* = 0.0202, respectively).

**Novel isoniazid resistance determinants explain resistance in phenotypically resistant strains with no known associated mutation.** Finally, we confirmed that mutations in candidate genes are relevant to clinical resistance. We reasoned that if our list of candidate genes plays a role in clinical resistance, we should detect an increment in the sensitivity values to predict isoniazid resistance not explained by available databases. We looked at a selected data set of strains obtained from the CRyPTIC consortium[6], enriched in isoniazid-resistant strains with no known resistance mutation (362 strains with known mutations, 82 with no known mutation). We found that the sensitivity of candidate genes was significantly greater than a random set of genes both for strains with no known mutations (sensitivity = 0.59, *p* = 0.019; Fig. 4a) and for strains harboring well-known resistance mutations (sensitivity = 0.46, *p* = 0.027; Fig. 4b and Supplementary Data 7). The result suggests that our list of genes is indeed involved in isoniazid resistance in one way or another. In both cases, any non-synonymous mutations in known resistance genes were also included as candidates, but they only contributed a small amount to the total sensitivity (Fig. 4a, b). By including rare mutations in our candidate genes list, we could increase global sensitivity from 93.1% to 94.6%, or from 97.1% to 98.9% if we omit genotypes with no clear prediction, further confirming that candidate genes in our analysis are relevant to isoniazid clinical resistance and could help explain uncommon resistant phenotypes.

The resistance mechanisms involved seem to be multiple and diverse. For instance, two of the genes with the most mutations in resistant strains are *dnaA* and *mutT4*, which are involved in DNA replication and may affect the acquisition of resistance by increasing mutation rate or indirect mechanisms such as decreasing expression levels of katG[24]. Other interesting genes in the candidate set are those encoding epoxide hydrolases (*ephD* and *ephF*), which are involved in detoxification following oxidative damage to lipids. Also, we find many putative candidate mutations in genes involved in transport such as *mmpL4* and *stp*, which can sometimes confer resistance to some antibiotics[25]. Finally, we find several genes with no known function that are accumulating many non-synonymous mutations in resistant strains. These results confirm that there are still many unknown factors affecting the evolution of antibiotic resistance in tuberculosis, even for a very well-characterized drug such as isoniazid, and highlight the need for systematic studies to uncover them.

## Discussion
In this work we show how functional genomics experiments have proven to be a very potent technique for finding resistance

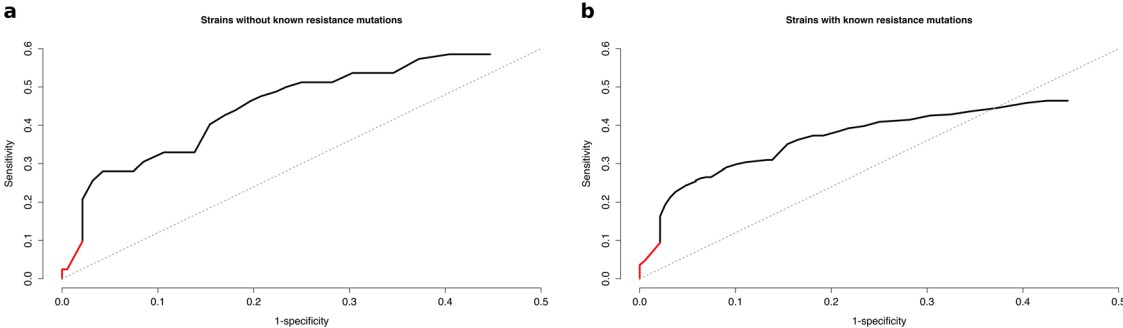

**Fig. 4 Novel isoniazid resistance determinants explain resistance in clinical strains. a** ROC curve for the data set of strains without known resistance mutations. **b** ROC curve for the data set of strains with known resistance mutations. The red portion of the curve corresponds to candidate mutations in known resistance genes. The black portion corresponds to candidate mutations in novel drug resistance-associated genes found in this study.

determinants in *M. tuberculosis*, as they allowed us to correctly identify five known isoniazid resistance regions as well as discover many other candidate resistance genes. Interestingly, out of these five previously known regions, only *katG* insertion mutants showed increased isoniazid resistance, whereas mutants for the other four regions had higher sensitivity instead. Thus, sensitivity-increasing genes can also act as resistance determinants depending on how mutations affect the protein or its expression levels. One example would be the *Rv2170* gene, which in our study increases sensitivity to isoniazid and has been shown to confer resistance when its expression is increased, because it encodes an acetyl-transferase that inactivates isoniazid[26]. Although we could not find insertion mutants for target isoniazid gene *inhA* due to its essentiality, non-essential genes in the mycolic acids biosynthesis pathway were functionally more associated with resistance than the rest of the genome, showing that the target pathway can be identified even when the target gene itself is essential. Functional genomics also pointed to redox metabolism as a resistance mechanism for isoniazid, confirming the usefulness of this technique in highlighting resistance determinants not directly tied to the drug's mode of action and in helping complete the resistance mutations catalog. Finally, combining functional genomics with phylogenetic data allowed us to pinpoint which regions and pathways were most important for resistance in clinical settings and to obtain a series of candidate novel resistance determinants. This last step is very important, it has been shown before that even the most common clinical mutations to isoniazid are difficult to recover in vitro[27]. Likewise, we find regions that have been shown to affect resistance in vitro but are irrelevant in vivo. For instance, the aforementioned Rv2170 has no phylogenetic association with resistance according to our results.

Many genes involved in cell wall biosynthesis, including the mycolic acids biosynthesis pathway, have been highlighted by both the functional and the phylogenetic approaches, even though many of them are not used to predict resistance (Fig. 3d). This fits in with the idea that resistance depends on more than the well-characterized diagnostic mutations, and that bacteria can also acquire low-effect resistance mutations or compensatory mutations. Given that isoniazid specifically targets this pathway, it is plausible that insertions in several genes in the pathway affect isoniazid sensitivity. However, as the cell wall is the first barrier of defense of the bacteria, it is difficult to determine whether these genes are important for isoniazid resistance exclusively or also for resistance to other antibiotics as well. For instance, our data show that inserting genes in the *mce1* operon increases isoniazid resistance. This operon is proposed to be involved in the transport of cell wall components and disrupting the operon is associated with the accumulation of free mycolic acids[28–30]. This

points to a role in cell wall remodeling and recycling and suggests that the operon can be involved in resistance to other antibiotics as well. Other genes such as those in the mycolic acid modification pathways are better candidates to affect isoniazid resistance exclusively. In any case, we expect many of the mutations in these genes to have low diagnostic value even if they actually contribute to resistance, which underlines the importance of systematic studies to understand antibiotic resistance.

Our analysis also highlighted genes involved in redox metabolism, which has previously been associated with isoniazid resistance[31]. This is in accordance with what is already known about the action of the antibiotic, as NADH is required for the formation of the isoniazid-NAD adduct. In addition, NADH and NADPH are necessary for the activity of two genes in the FAS system, *inhA* and *fabG1*, which we know to be isoniazid resistance genes. However, as most of the oxidative metabolism genes we found to be associated with resistance were not directly involved in any of these pathways we concluded that this association depends on NADH homeostasis. Previous findings have suggested that NADH dehydrogenase gene *ndh* harbors putative resistance mutations[14] and our results showed that many similar genes could have such mutations as well. For instance, we found that genes encoding NADH dehydrogenases present in the electron transport chain, such as several genes in the *nuo* operon and *ndhA*, increased isoniazid resistance when mutated. It has already been shown that inhibiting the action of these genes increases intracellular levels of NADH[32], and that mutations in the NADH dehydrogenase Ndh (isoform to NdhA) lead to higher NADH levels and confer isoniazid resistance, maybe by preventing inhibition of the InhA enzyme[31]. Indeed, we found that mutations in *nuoJ*, *ndhA*, and *ndh* have phylogenetic association with resistance, further confirming that NADH homeostasis plays a clinically relevant role in the evolution of isoniazid resistance. NADH can also affect resistance indirectly, as several stress responses specifically use the NADH : NAD+ ratio as a trigger[33]. In many cases, these responses provide protection against antibiotic stress as well and are triggered as a result of exposure to the antibiotic[32]. We identified several NADH sensors such as *regX3*, *dosT*, and *pknG*, which were functionally associated with isoniazid resistance. In addition to redox sensors, we also found other genes related to detoxification that were associated with resistance. For instance, insertions in genes involved in mycothiol biosynthesis produced a more resistant phenotype.

Once we understand which pathways are involved in isoniazid resistance, we can start exploring them as targets for future regimens. Many sensitivity-increasing features in the cell wall biosynthesis pathway show increased expression after exposure to isoniazid[34]. The insertion probably compromises the strength and integrity of the cell wall and makes those genes ideal targets for

adjuvants that help potentiate the action of the antibiotics or even for new or repurposed therapies. For instance, peptidoglycan biosynthesis gene *ponA1* appears to be the target of the repurposed ceftazidime-avibactam combination[35]. We can also take advantage of the role of redox homeostasis in the action of isoniazid to design treatments and strategies to enhance the performance of the antibiotic. In a recent paper, Flentie et al.[7] reported a new compound, C10, which appears to revert the resistant phenotype of *katG* mutants and thus prevents the selection of isoniazid-resistant variants. The compound had been specifically selected to block tolerance to oxidative stress and it was shown to both increase sensitivity to isoniazid and promote the expression of energy metabolism-related genes. Here we have provided experimental evidence of which specific genes can produce a particular resistance phenotype and which of those are relevant in clinical environments. We can now use this experimental framework and extrapolate it to other antibiotics, including newly licensed drugs such as bedaquiline and delamanid to design better treatments with them.

A very powerful feature of functional genomics is that it can be applied comparatively across strains and antibiotics. For instance, functional genomics showed that small genetic differences between characterized strains could be linked to differences in relative importance or essentiality of particular genes and in the way a strain acquires antibiotic resistance[36]. When we use functional genomics with different antibiotics, we can find common patterns of resistance across drugs, such as the existence of an intrinsic resistome[15,37] and instances of cross-resistance. In our data, we found that the F420 biosynthesis gene *fbiA-C*, which are associated with delamanid resistance[38], also confers isoniazid resistance when insertionally inactivated. These cross-resistance patterns are important, as they inform us of the likelihood that the bacteria can develop resistance to two antibiotics that are administered in combination or sequentially, which will ultimately impact treatment success.

By combining a functional and phylogenetic approach, we have shown that our candidate resistance determinants could increase sensitivity up to 2%. Although the impact at the population level seems minimal, the impact for the patient is important as isoniazid resistance is a major determinant for adverse clinical outcomes[39]. Furthermore, genome-wide association studies were unable to reveal any of these targets when thousand of genotypes–phenotypes[6] or detailed MIC measurements[40] were implemented. In our approach, we are able to identify novel regions that are involved in isoniazid resistance, as they increase sensitivity in sets of phenotypically resistant strains with no known causing mutation. Nevertheless, the use of those regions to diagnose isoniazid resistance leads to a decline in specificity, suggesting that not all mutations in the target genes are involved in resistance. Furthermore, mutations in some of the regions are predictive of resistance in strains with very well-characterized isoniazid resistance mutations, suggesting that in some cases they can act either as compensatory mutations or early low level, facilitating resistance mutations that preceded the well-characterized isoniazid resistance mutations.

In conclusion, functional genomics is a powerful tool for detecting hundreds of genomic determinants that can modify antibiotic resistance, but we need a clinical readout to determine which of the genes have real relevance to the evolution of antibiotic resistance and the emergence of clinical resistance. Here we have shown how a systematic approach combining insertion mutants on a genomic scale with phylogenetic association of clinical mutations can reliably detect important resistance features, uncover new resistance candidates, and highlight relevant pathways and potential cross-resistances, in this case those most related to the mode of action for the first-line antibiotic isoniazid.

We also provide an example of how a thorough understanding of the genetic architecture of isoniazid resistance can help us to prevent its emergence. The approach we describe can be used as a blueprint for studying the genetics of resistance to other antibiotics or describing lineage-specific differences, particularly to provide much-needed knowledge regarding resistance to new antibiotics.

## Methods

**Strains, media, and culture conditions**. We used *M. tuberculosis* strain H37Rv kindly supplied by Darío García de Viedma. Bacteria were cultured at 37 °C in Middlebrook 7H9 supplemented with 10% ADC (both from BD) and 0.05% Tween 80 (Difco) for liquid cultures and in Middlebrook 7H10 supplemented with 10% OADC (both from BD) for solid cultures. All experiments were conducted in a BSL3 laboratory using a biosafety cabinet.

**Mutant pool generation and selection experiment**. We generated a mutant pool using the protocol by Long et al.[8]. Briefly, we collected 100 mL *M. tuberculosis* H37Rv culture and washed twice with mycobacteriophage (MP) buffer to remove the Tween. We then transduced the bacteria with $10^{11}$ pfu phiMycomarT7 for 20 h in a total volume of 10 mL. Afterwards, we pelleted the cells and washed away excess phage twice with PBS-Tween 80. Finally, we plated the transduced bacteria in three 25 × 25 square plates (Corning) containing Middlebrook 7H10 media supplemented with OADC, 0.05% Tween 80, and 20 μg/mL kanamycin (Panreac). After 3 weeks, mutant bacteria were scraped from the agar, homogenized in liquid media, and stored at −80 °C.

For each experiment, we used a starter culture of the pool at OD 0.8–1.0 to inoculate two 100 mL roller bottles in parallel with ~$10^7$ bacteria each. One of the bottles contained either 0.18 or 0.20 μg/mL isoniazid (Panreac), whereas the other contained just plain media and served as a control. We allowed the bacteria to grow for about 13 generations and stored the final populations at −80 °C.

**DNA extraction**. Mycobacterial cultures were pelleted by centrifugation and resuspended in 500 μL TE buffer. Following inactivation by heat at 80 °C for 1 h, lysozyme (50 μL of a 10 mg/mL stock) was added and samples were incubated overnight at 37 °C. Then, 50 μL of proteinase K (10 mg/mL stock solution) were added, incubating for 1 h at 60 °C with shaking in a thermomixer. After this, 100 μL 5 M NaCl and 100 μL 10% CTAB were mixed by inverting, samples were frozen for 15 min at −80 °C, and re-incubated at 60 °C for 15 min with shaking. Once cooled, 700 μL chloroform–isoamyl alcohol (24 : 1) were mixed in, yielding a white, homogenous solution. Samples were centrifuged, transferred into 700 μL cold isopropanol, and left at −20 °C overnight. Then, they were pelleted and washed with 70% ethanol and dried in a speed vacuum concentrator for 10 min. Finally, DNA was resuspended in 50 μL TE and its concentration was determined with a QuBit 3.0 Fluorometer (Thermo Fisher Scientific). Also, the amount of contaminating phage DNA was estimated by PCR, to ensure it was low and would not affect sequencing results.

**Library preparation and sequencing**. We followed the protocol described by Long et al.[8] with some modifications. After extraction, samples were quantified using a QuBit 3.0 Fluorometer (Thermo Fisher Scientific). Then, 50 μl of each DNA sample were transferred to Covaris tubes, centrifuged, and fragmented to an approximate size of 550 bp using standard settings (Illumina TruSeq Library Prep Reference Guide). Samples were size-selected using NucleoMag NGS Clean-up and Size Select (Macherey-Nagel) to 50 μL final volume. The DNA end-repair was performed using NEBNext End Repair Module (New England BioLabs) and dA-tailing was achieved with NEBNext dA-tailing Module (New England BioLabs), both as per the manufacturer's instructions. Next, a stock of barcoded adapters was prepared by mixing 20 μl of 50 μM oligonucleotides (detailed in Supplementary Table 1) in a final concentration of 2 μM $MgCl_2$, heating to 93 °C for 10 min, and reducing the temperature by 3 °C/cycle over 2 h until reaching 20 °C. These double-stranded adapters were then ligated to the purified dA-tailed DNA using T4 DNA ligase from NEBNext Quick Ligation Module (New England BioLabs).

After another purification step, we performed a PCR to selectively amplify the transposon-chromosomal junctions using a pair of primers specific to the end of the transposon and ligated adapter (Supplementary Table 1) with the following parameters: initial denaturation at 98 °C for 5 min, 20 cycles of denaturation at 98 °C for 20 s, annealing at 65 °C for 15 s, and extension at 72 °C for 30 s, with a final extension at 72 °C for 3 min. We size-selected ~500 bp products using magnetics beads and a standard eight-cycle indexing PCR introduced the Illumina indexes required for sequencing. Final libraries were validated on a Bioanalyzer DNA chip (Agilent Technologies) to verify size and then quantified again using Qubit.

Libraries were sequenced on the Illumina NextSeq 550 platform using the High Output v2 kit (150 cycles), producing an average of 35 million raw paired reads per sample with a good quality distribution.

**Bioinformatic analysis**. Quality control of sequencing files was performed using FASTQC, after which they underwent quality trimming by PRINSEQ. The selected criteria for keeping sequences was a mean Phred quality score of 20 in a 20 bp sliding window. Next we processed the cleaned sequences by means of a custom Python script that served two purposes: for every read pair, it first scanned the beginning of the forward read looking for a "TGTTA" motif that marked the start of the transposon insertion and cut the sequence at the TA site; second, it looked for the random barcode in the reverse read, cut it from the sequence, and appended it to the header as a comment if it passed a structure check.

We then mapped the reads to the *M. tuberculosis* H37Rv reference genome (NC_000962.3) using the Burrows-Wheeler Aligner (BWA) with default parameters but keeping header comments in the resulting SAM files. It was important to detect PCR duplicates and remove them to correctly estimate the proportion of each mutant in the original pool. This step was again performed by a custom Python script that used the barcode, strand, mapping coordinate, and fragment length information to define unique reads. After removing PCR duplicates, the insertion count final list for each sample was generated, with coordinates determined by the mapping point of the "TA" site. At this point, we were able to determine a ~78% insertion density and thus the high quality of our mutant pool.

We developed our own pipeline to analyze the TnSeq data. Our idea was to determine which genes consistently alter isoniazid resistance and current approaches work better when the size of the effect is large. First, we normalized insertion counts by 40% trimmed mean and generated a T0 library that was sampled 1000 times to obtain z-scores that represented the standardized deviation of each site from its expected insertion count. We also generated two new libraries, R1 and R2, by subtracting z-scores from controls to their treatment's counterparts. A sliding window analysis was designed to evaluate the significance of inserted genomic features with the aim of detecting zones with equally consistent insertion changes. Each window containing between six and ten "TA" sites underwent a Wilcoxon test that deemed whether the region was inserted more or less than expected. We only considered windows that fell within the limits of annotated genomic features and eliminated the ones that spanned more than one. In terms of fitness, a window above the expected insertion level was defined as an increased resistance window and one below the expected insertion level was considered an increased sensitivity window. A multitest correction was applied to all p-values after the analysis, obtaining a final q-value for each window. Then we established a call for each "TA" site by judging the windows in which it appeared. If half of its windows got a significant call, that call was applied to the "TA" site, otherwise it was left as nonsignificant. Uninserted "TA" sites in the T0 libraries were not called and were annotated as "not evaluated." Finally, "TA" sites were assigned to genomic features to give them a call. Features containing at least 70% of a certain significant call were given that same call if their median z-score was either positive or negative and their ranking among all features was above or below 50%, respectively. We selected all significant calls from either of the two concentrations tested and a final call for each gene and intergenic region was obtained following this pipeline. Scripts necessary to perform the analyses are available at https://gitlab.com/tbgenomicsunit.

To determine how reproducible our results were, we compared them with published data from Xu et al.[15]. We found approximately five times more resistance-altering features than they did, suggesting that our results might provide a more detailed picture of the genetic architecture of isoniazid resistance. This is probably due to a combination of stronger selective conditions, an increased number of inserted sites and the fact that our analytical method tests for consistency independently of the size of the effect. Coincidence between our and their sets of resistance-altering genes was significantly higher than expected ($\chi^2$-test, $\chi = 297.28$, $p < 0.0001$), with particularly strong association in the direction of the effect (Supplementary Fig. 3a). Finally, we checked whether an effect in the same direction was found for all our resistance-altering regions even if it was not significant. Indeed, our resistance-increasing features had significantly higher resistance than nonsignificant ones (Wilcoxon test, $z = 8.93$, $p < 0.0001$), whereas sensitivity-increasing features showed lower resistance (Wilcoxon test, $z = 11.54$, $p < 0.0001$; Supplementary Fig. 3b).

**Clinical data set phylogeny and ancestral state reconstruction**. A 4763-strain data set consisting of different worldwide clinical isolates was constructed from publicly available databases. We downloaded all FASTQ files from NCBI using their fastq-dump tool, mapped them to a predicted *M. tuberculosis* ancestor reference and called SNPs using VarScan 2. An alignment of all homozygous variable positions among isolates was generated and a phylogenetic tree was constructed using FastTree 2.1.

We then proceeded to reconstruct the ancestral state of every polymorphism using PAUP 4.0a158 with a custom weight matrix that punished reversions with a 10× multiplier (see Supplementary Note 1 for the assumptions block including this matrix). As we had more positions than the program could compute at a time, we had to split the alignment into four 60K-SNP pieces before building the NEXUS files as input for the program. We obtained a list of changes associated with the tree's nodes and this output was parsed using a custom Python script, yielding a summary table of changes that was further processed to add more information about each change. The final table contains for each SNP event the tree node in

which it occurs, associated genomic feature, translational impact, homoplasy, and antibiotic resistance details (Supplementary Data 4).

**Phylogenetic association test**. We used R to perform an association test linking particular SNPs occurring in clinical settings to antibiotic resistance. We began by defining susceptible and resistant tree branches according to the absence or presence, respectively, of an antibiotic-resistance-associated mutation from a high-confidence resistance mutations list based on PhyResSE[22] and ReSeqTB (Supplementary Data 5). Along with isoniazid we considered all first-line antibiotics, because although isoniazid resistance tends to appear first[41], resistance mutations are not always detected. Furthermore, resistant strains can accumulate non-specific or low-level rare mutations and we are also interested in those mutations. In any case, we repeated the analysis only with isoniazid resistance mutations and the results were very similar. After we determined "resistant" and "sensitive" subtrees of the phylogeny, we eliminated all diagnostic mutations and calculated the number of all mutations, non-synonymous mutations and homoplasies for each genomic feature in resistant subtrees. We then determined the expected number of mutations in each category by resampling the assigned branch for each mutation 10,000 times and recalculating the numbers for each category. We determined that one particular genomic feature was phylogenetically associated with resistance if it ranked higher than 9500 of the 10,000 samplings in any of the three categories.

We used the entire list of mutations for all first and second-line antibiotics for several reasons as follows: (i) we expect most strains resistant to other antibiotics to be resistant to isoniazid as well, as not all isoniazid resistance mutations are known and resistance mutations in tuberculosis tend to appear in a stepwise fashion with isoniazid resistance mutations being one of the first; (ii) even in the cases where no proper isoniazid resistance mutation has occurred, other low-level resistance mutations may have been acquired and they are also relevant to the evolution of resistance; and (iii) as mutations that confer resistance to different antibiotics are highly correlated due to the nature of the treatment, it is very difficult to disentangle one from the other and it is better to study resistance as a whole independently of the specific antibiotic. All diagnostic mutations used to mark the onset of resistance in the phylogeny were subsequently eliminated from analysis.

**Resistance prediction in the clinical data set**. We used 444 isoniazid-resistant strains from the Cryptic data set[6] (Supplementary Data 8): 362 with typical isoniazid resistance mutations and 82 without any known isoniazid resistance mutations. We analyzed the raw sequence data using our lab's validated pipeline (available at https://gitlab.com/tbgenomicsunit/ThePipeline) and determined all the single-nucleotide mutations above a 10% frequency for each strain in the sample. We centered our analysis on "rare" mutations (mutations appearing in only one to three strains) and all diagnostic resistance mutations were excluded. We first determined the number of strains in each group (resistant with typical mutations or resistant with no clear mutations) that contained at least one mutation in any of the candidate genes and then we compared this number to the results we obtained using 1000 random subsets of genes with the same number of features as our candidates list (PE/PPE protein families, phage, and repetitive sequences excluded)[42]. The relative position of the sensitivity obtained with our candidate list revealed how relevant those genes are to clinical resistance. To build the receiver operating characteristic (ROC) curve, we similarly analyzed 188 randomly sampled pan-susceptible strains from the CRypTIC consortium and added each gene sequentially in descending order of specificity. When two genes had the same specificity, the one with the highest sensitivity took precedence.

**Candidates validation using BCG mutants**. We selected 30 BCG Danish mutants (24 candidate genes plus 6 controls) from the BCCM/ITM Mycobacteria Collection, which were regrown and shipped in 7H11 solid medium. Upon reception, bacteria were collected and further amplified in 7H9 liquid medium to generate stocks for MIC determination experiments. We set up 96-well microtitre plates (using 7H9-OADC) with 2-fold serial dilutions of isoniazid, with concentrations ranging from 0.015 to 8 μg/mL and leaving 2 wells without antibiotic to have a growth control. BCG mutants were inoculated by rows, adding $10^4$ bacteria per well, along with a wild-type BCG Danish strain as internal control in each plate. After a 7-day incubation at 37 °C, 20 μL of 0.02% resazurin were added to each well and plates were allowed to incubate for 2 more days. At the 24 and 48 h time points, resazurin color change was visually assessed and a 50 μL aliquot was inactivated with 50 μL of 4% paraformaldehyde and placed in a black plate for fluorescence reading. Resazurin reduction was assessed by measuring fluorescence in a Tecan Infinite M Plex plate reader, allowing for a precise estimation of IC50 and IC90 values. Briefly, we subtracted the negative controls and re-calculated the fluorescence values as relatives to their respective growth control. We determined the slope of the steepest part of the inhibition curve and used that estimation to determine the concentrations at which inhibition was exactly 50% and 90% relative to the growth control.

**Statistics and reproducibility**. We performed all statistical analyses using R v3.3.3 and publicly available databases (cited in the main text). The specific test performed in each case (Fisher's exact test, Wilcoxon test, or custom) is indicated throughout the text. The TnSeq selection experiment was performed twice and the

same pool was tested in two independent blocks, in each block we seeded one control and one experimental culture from the same starter culture. Depth of sequencing for the TnSeq experiment was designed so that there would be around 50 reads per insertion, to ensure adequate coverage of all insertion sites. Experimental cultures were inoculated with 10 million bacteria, ensuring that on average each insertion mutant would have 100 copies. We allowed bacteria to grow for 13 generations, because that is enough to see differences in growth for the different mutants. We compared with similar results from other groups and found good agreement. We tested 24 insertion mutants for candidate genes for confirmation, comparing them with 6 control mutants. Insertion mutants were tested in randomized blocks and with a quality control strain.

All analyses can be reproduced using data sets and code available at https://gitlab.com/tbgenomicsunit.

**Reporting summary**. Further information on research design is available in the Nature Research Reporting Summary linked to this article.

## Data availability
TnSeq FASTQ files are available at the European Nucleotide Archive with project number PRJEB38844. Strains available under request. Supplementary Data 1–4 are also available at https://gitlab.com/tbgenomicsunit/tnseq-pipeline. Source data for the main figures is available in Supplementary Data 9.

## Code availability
Custom code used in this study in the form of Python scripts are available at https://gitlab.com/tbgenomicsunit/tnseq-pipeline (https://doi.org/10.5281/zenodo.5575540)[43]. The code is available under Creative Commons Attribution 4.0 International.

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

## Acknowledgements
This project has received funding from the European Research Council (ERC) under the European Union's Horizon 2020 research and innovation programs 638553 (TB-ACCEL-ERATE), SAF2016-77346-R from Ministerio de Economía y Competitividad (Spanish Government) and AICO/2018/113 from Generalitat Valenciana (to I.C.). M.M.-M. is recipient of a FPI grant from the Ministerio de Economía y Competitividad (Spanish Government, code BES-2017-079656). V.F. was recipient of a post-doctoral research grant from the Ministerio de Economía y Competitividad (Spanish Government, code FPDI-2013-18757). Action co-financed by the European Union through the Operational Program of European Regional Development Fund (ERDF) of Valencia Region (Spain) 2014-2020.

## Author contributions

V.F. and I.C. designed the experiments, analyzed the data, and wrote the paper. In addition, V.F. performed the BSL3 experiments. M.M.-M. performed the bioinformatic analysis and performed additional BSL3 experiments. A.C.-O. performed additional bioinformatic analysis. L.V. contributed technical assistance to the BSL3 experiments. M.T.-P. generated the TnSeq libraries. All authors contributed to the writing of the manuscript.

## Competing interests

The authors declare no competing interests.
