## [Transparent Peer Review File · Communications Biology]

Reviewers' comments:

Reviewer #1 (Remarks to the Author):

The study by Furio and co-authors focuses on the evaluation of genetic determinants of isoniazid resistance by creating a library of transposon mutants, selecting for isoniazid resistance and performing a deep functional genomics and phylogenetic analysis.

In my opinion, the study is well designed and presents conclusion of potential relevance to researchers and clinicians interested in understanding and diagnosing isoniazid resistance.

The study results in large amounts of information that would be difficult to cover in its entirety, but I believe the authors made a good effort to highlight the most important findings. I found only minor issues that given the scope of the manuscript should not preclude the manuscript from publication:

1. Fig1c shows a large difference in share counts of the top 100 sites between experiment 1 and 2 with isoniazid. The methods also indicate the added isoniazid was either 0.18 or 0.2 ug/ml. Could the differences be explained by the small difference in INH concentration?

2. It is unclear why the authors indicate that only insertions in *katG* showed increased isoniazid resistance while other insertions resulted in increased sensitivity (line 365-366). However, table 1 and text elsewhere (i.e. line 339-341 (*dnaA* and *mutA*); and 395-396 (*mce1*)) indicate that other mutations were also associated with increased resistance.

3. Authors should reference a recently published manuscript (Hicks et al, PLOS Pathogens) in which mutations in *dnaA* are shown to increase isoniazid resistance in *M. tuberculosis* by reducing expression of *katG*.

Reviewer #2 (Remarks to the Author):

The manuscript by Furio et al presents an analysis of genes involved in isoniazid resistance by combining TnSeq data and phylogenetic analysis of SNPs in a global collection of clinical isolates. First, 555 genes and 126 intergenic regions are identified in a TnSeq library generated in H37Rv and selected on isoniazid (sub-inhibitory concentration). Then these candidate genes are checked for excess polymorphisms associated with drug resistance. Pathway analysis suggests that redox metabolism and cell-wall biosynthesis pathways are enriched. However, it should be noted that the relevance of redox metabolism and cell-wall synthesis to the INH mechanism of action has been known for decades.

The TnSeq study is well-done, with two independent replicate libraries that are well-saturated (with insertions at ~78% of TA sites). The authors observe that several genes expected to play a role in the INH mechanism of action or resistance appear conditionally essential, including *KatG*, *InhA* promoter, NADH dehydrogenases, etc. Furthermore, these have an excess of polymorphisms in clinical isolates. The authors demonstrate that their results are in concordance with a previously published TnSeq study of H37Rv treated with INH (Xu et al, 2017).

One limitation however, as the authors acknowledge, is that TnSeq is based

on all-or-nothing disruption of function, and cannot assess the role of SNPs (e.g. non-synonymous substitutions). So the signals from the two experiments (TnSeq and GWAS) cannot be expected to always overlap. Nonetheless, as a post-hoc filter, looking at genes associated with SNPs in drug-resistant strains is a useful way to evaluate and prioritize significant genes from TnSeq analysis.

Although the Introduction discusses that fact that there is a significant fraction of strains resistant to some drugs that have unexplained resistance in genome databases, and polymorphisms in some genes known to be involved peripherally have very weak statistical associations, the Intro overlooks the potential lineage-dependence of effects of mutations on resistance, which could be a major reason why some allelic signals are weak; i.e. SNPs in a locus might impact sensitivity for some strains and not others. In fact, the MIC for many drugs has been shown to vary across clinical isolates, suggesting that epigenetic effects could play a role. In this light, the fact that the TnSeq experiment was performed with a library made only in the H37Rv reference strain is a limitation, which the authors might want to address.

The manuscript is speculative in places. For example, the discussion of "probable mechanisms" such as efflux in lines 292-298. While true, the authors should be more specific about which of their data directly supports this. Similarly, in lines 300-308, the authors conclude that cell wall biosynthesis pathways are important to INH sensitivity. But the first sentence says only 1 cell-wall-related gene (*fabG1*) was observed to be significant in their TnSeq data. Thus it seems overblown to say this is a conclusion from their experiment.

In lines 310-316, it is observed that 8 of 42 redox genes showed significant effects. However, disruption of 7 out of 8 of the genes showed increased resistance. It would help if the authors could interpret whether increased resistance or increased sensitivity would be expected, given previous results on the importance of redox homeostasis on INH. Also, they point out that genes phylogenetically associated with resistance as a whole were not enriched in redox genes. So it seems counter-intuitive to conclude that "These results confirm that redox metabolism plays a clinically relevant role in the evolution of isoniazid resistance".

One of the main limitations of the manuscript is that none of the candidate genes (e.g. *dnaA*, *Rv2170*) is validated. The fact that a gene is significant in a TnSeq experiment only suggests but does not prove that disruption affects the sensitivity to a drug; one has to make the mutant and show a shift in MIC, or a drug-specific effect on the growth curve.

The statistical model used in this paper (Mann-Whitney test) might be too "liberal"; it seems implausible that 555 genes (~15% of genes in the genome) actually influence sensitivity to INH when disrupted. Some observations mentioned are difficult to rationalize, such as the implication that disruptions of *fbiA-C* would affect INH sensitivity.

For the analysis of polymorphisms in the clinical isolates, It appears that the "resistance" phenotype is defined to include "resistance to any drug", instead of specifically INH resistance. The authors comment that their results would be similar if they had specifically focused on INH-resistant strains. If that is true, I would strongly recommend they revise their analysis to use only INH-resistance as a phenotype. Otherwise, they risk confounding polymorphisms at loci associated with resistance to other drugs (since there is a lot of co-resistance in these databases). This could lead to false positives in their analysis (i.e. genes not specifically related to INH resistance).

Also, why do the authors choose to use a binary definition of resistance (only R and S)? It might have been more informative to take the MIC into account quantitatively (MIC is available some but not all the strains in these databases, and they could have used a subset of strains). This is especially relevant for INH, because it is well-known that some mutations are associated with high-level resistance, and others with low-level resistance.

Specific comments:

Fig 1b: Why is there such a big difference for growth curves for INH treatment between the 2 experiments (libraries)? Can the authors include a scatter plot (as a Supplemental Figure) showing the correlation between mean insertion counts in the genes between the two replicates? (i.e. to assess reproducibility)

line 221: Why was the binomial test used instead of the hypergeometric test (Fisher's Exact Test), which is more commonly used for this purpose in the literature on pathway analysis? It could make a difference on which pathways are significant, especially with pathways where only a few genes are involved.

Fig 4: How many strains out of ~4,762 actually have *unexplained* INH resistance?

Reviewer #3 (Remarks to the Author):

In this manuscript, the authors use transposon sequencing to comprehensively assess genes associated with susceptibility and resistance of Mycobacterium tuberculosis to the first line drug isoniazid. To lend support for a role of the identified genes in isoniazid resistance the authors query a panel of genome sequences of characterized resistant isolates for variations in the respective genes. From the findings presented, the authors infer a new set of genes that are associated with resistance and susceptibility to isoniazid.

Major points:

While this paper presents some potentially important findings relating to novel molecular mechanisms for isoniazid resistance, additional rigor is essential to confirm associations between genotype and phenotype and to assess mechanisms at play. The work could be improved with a more thorough analysis of the data, and with follow up experiments to confirm association of a few of the top

candidate genes in the analysis with isoniazid resistance. Further, use of a single library for the analysis is not viewed as rigorous. It is difficult to follow the details of the Tnseq analysis.

Display figures are very challenging to understand, in part, due to the extremely small font size that was used, but also due to the limited description of the results in the text (especially Figure 3 and 4).

Minor points:

line 36, change "curation" to "cure"

line 43 change "resistant" to "resistance"

line 61 change "helper" to "companion"

line 69 change "resistance" to "resistant"

line 128 change "inserted" to "insertion"

line 133 change "it only allows" to "they only allow"

line 135 change "be inserted" to "tolerate insertion"

line 148 change "frequencies" to "abundance"

lines 150-153, awkward wording, perhaps try "Isoniazid-containing cultures show strong enrichment of a fraction of insertions indicating a selective advantage relative to the bulk of the population."

line 156 change "inserted" to "insertion"

line 157 change "independently" to "independent"

line 304 change "be inserted" to "tolerate insertion"

line 416 change "inserted" to "mutated"

line 428 change "inserting" to "insertions in"

line 458 change "inserted" to "insertionally inactivated"

Reviewers' comments:

Reviewer #1 (Remarks to the Author):

The study by Furio and co-authors focuses on the evaluation of genetic determinants of isoniazid resistance by creating a library of transposon mutants, selecting for isoniazid resistance and performing a deep functional genomics and phylogenetic analysis.

In my opinion, the study is well designed and presents conclusion of potential relevance to researchers and clinicians interested in understanding and diagnosing isoniazid resistance.

The study results in large amounts of information that would be difficult to cover in its entirety, but I believe the authors made a good effort to highlight the most important findings. I found only minor issues that given the scope of the manuscript should not preclude the manuscript from publication:

1. Fig1c shows a large difference in share counts of the top 100 sites between experiment 1 and 2 with isoniazid. The methods also indicate the added isoniazid was either 0.18 or 0.2 ug/ml. Could the differences be explained by the small difference in INH concentration?

As the concentrations used in the experiment are very close to the MIC and the inhibition curve is very steep there can be big differences in the degree of inhibition (and by extension on the strength of selection) with relatively small changes in isoniazid concentration. In this case, we would precisely expect more extreme differences with the higher concentration even if the difference is small.

2. It is unclear why the authors indicate that only insertions in *katG* showed increased isoniazid resistance while other insertions resulted in increased sensitivity (line 365-366). However, table 1 and text elsewhere (i.e. line 339-341 (*dnaA* and *mutA*); and 395-396 (*mce1*)) indicate that other mutations were also associated with increased resistance.

In this case, we were indicating that out of the already known genes for isoniazid resistance only *katG* showed increased resistance when inserted. We have made changes at the beginning of the Discussion section to reflect this.

3. Authors should reference a recently published manuscript (Hicks et al, PLOS Pathogens) in which mutations in *dnaA* are shown to increase isoniazid resistance in *M. tuberculosis* by reducing expression of *katG*.

Thank you for bringing this reference to our attention, we have included it when mentioning *dnaA* at the end of the Results section of the manuscript.

Reviewer #2 (Remarks to the Author):

The manuscript by Furio et al presents an analysis of genes involved in isoniazid resistance by combining TnSeq data and phylogenetic analysis of SNPs in a global collection of clinical

isolates. First, 555 genes and 126 intergenic regions are identified in a TnSeq library generated in H37Rv and selected on isoniazid (sub-inhibitory concentration). Then these candidate genes are checked for excess polymorphisms associated with drug resistance. Pathway analysis suggests that redox metabolism and cell-wall biosynthesis pathways are enriched. However, it should be noted that the relevance of redox metabolism and cell-wall synthesis to the INH mechanism of action has been known for decades.

While it is true that these metabolic pathways have been known to be relevant for some time, here we put the emphasis on redox metabolism and cell-wall biosynthesis as evolutionary relevant resistance mechanisms. Additionally, this paper aims to work as a blueprint on how functional genomics can be used to understand resistance to less well-studied antibiotics. Thus, finding additional evidence for known resistance mechanisms is not a drawback in this case but a net positive.

The TnSeq study is well-done, with two independent replicate libraries that are well-saturated (with insertions at ~78% of TA sites). The authors observe that several genes expected to play a role in the INH mechanism of action or resistance appear conditionally essential, including KatG, InhA promoter, NADH dehydrogenases, etc. Furthermore, these have an excess of polymorphisms in clinical isolates. The authors demonstrate that their results are in concordance with a previously published TnSeq study of H37Rv treated with INH (Xu et al, 2017).

One limitation however, as the authors acknowledge, is that TnSeq is based on all-or-nothing disruption of function, and cannot assess the role of SNPs (e.g. non-synonymous substitutions). So the signals from the two experiments (TnSeq and GWAS) cannot be expected to always overlap. Nonetheless, as a post-hoc filter, looking at genes associated with SNPs in drug-resistant strains is a useful way to evaluate and prioritize significant genes from TnSeq analysis.

Although the Introduction discusses that fact that there is a significant fraction of strains resistant to some drugs that have unexplained resistance in genome databases, and polymorphisms in some genes known to be involved peripherally have very weak statistical associations, the Intro overlooks the potential lineage-dependence of effects of mutations on resistance, which could be a major reason why some allelic signals are weak; i.e. SNPs in a locus might impact sensitivity for some strains and not others. In fact, the MIC for many drugs has been shown to vary across clinical isolates, suggesting that epigenetic effects could play a role. In this light, the fact that the TnSeq experiment was performed with a library made only in the H37Rv reference strain is a limitation, which the authors might want to address.

This is an interesting point. Although looking at lineage-specific differences might indeed be worthwhile, here we are looking at common mechanisms for all tuberculosis lineages and use complementary approaches to accomplish that. As we used the phylogenetic association and validated the candidate genes in BCG we are rather confident that our results stand independently of genetic background. Lineage-specific differences would be a very interesting topic for future work. We have made a reference about lineage-specific differences as an interesting avenue for future work at the end of the discussion section.

The manuscript is speculative in places. For example, the discussion of "probable mechanisms" such as efflux in lines 292-298. While true, the authors should be more specific about which of their data directly supports this.

In the referenced lines, we are giving examples of mechanisms of resistance present in our candidate list and we are using publicly available information from the H37Rv reference genome annotation as stated in the Methods section. We have added some specific examples from our candidates for each listed mechanism plus we have referenced Table 1.

Similarly, in lines 300-308, the authors conclude that cell wall biosynthesis pathways are important to INH sensitivity. But the first sentence says only 1 cell-wall-related gene (fabG1) was observed to be significant in their TnSeq data. Thus it seems overblown to say this is a conclusion from their experiment.

The conclusion that cell wall biosynthesis pathways are important to INH sensitivity comes from pathway analysis of both TnSeq data and phylogenetic association data and that is made explicit in the text. However, it is possible that the combined evidence that we are using to extract this conclusion is too spread throughout the document. We have decided to reference back to the TnSeq pathway analysis results and Figure 3d where all these results are summarized.

In lines 310-316, it is observed that 8 of 42 redox genes showed significant effects. However, disruption of 7 out of 8 of the genes showed increased resistance. It would help if the authors could interpret whether increased resistance or increased sensitivity would be expected, given previous results on the importance of redox homeostasis on INH

What we state in these lines is that 8 out of 42 candidate genes were phylogenetically and functionally associated with resistance, and 7 out of those 8 showed increased resistance. The expectation of increased or decreased resistance after a disruption would depend on the specific gene, and the effect of particular mutations would depend on the mutation. With the current data it is very difficult to predict.

Also, they point out that genes phylogenetically associated with resistance as a whole were not enriched in redox genes. So it seems counter-intuitive to conclude that "These results confirm that redox metabolism plays a clinically relevant role in the evolution of isoniazid resistance".

We do not think it is counter-intuitive at all. We must bear in mind that genes phylogenetically associated with resistance might not have a direct relationship with resistance. For instance, rpoC mutations are strongly associated with resistance but they do not confer a resistant phenotype. Similarly, many genes can tend to accumulate mutations in resistant strains for instance to compensate for the cost of resistance. Thus, it is logical to find that a particular type of genes (i.e. redox genes) is enriched within the subgroups of isoniazid resistant candidate genes and not in the wider group of phylogenetically-associated genes.

One of the main limitations of the manuscript is that none of the candidate genes (e.g. *dnaA*, Rv2170) is validated. The fact that a gene is significant in a TnSeq experiment only suggests but does not prove that disruption affects the sensitivity to a drug; one has to make the mutant and show a shift in MIC, or a drug-specific effect on the growth curve.

Thanks to the reviewer for this comment. As they may be aware, obtaining knock-out mutants and testing them is time-consuming in *Mtb*. We have had access to a collection of KO mutants for some of the genes of interest in a BCG background (covering over half the list). For those KO we have determined the MICs and we have found that mutants that were expected to increase their resistance had a significantly higher MIC than either those that were expected to decrease it or those that were expected to show no change. We have included a new file as Supplementary Material with details about this experiment and its results for this resubmission.

The statistical model used in this paper (Mann-Whitney test) might be too "liberal"; it seems implausible that 555 genes (~15% of genes in the genome) actually influence sensitivity to INH when disrupted. Some observations mentioned are difficult to rationalize, such as the implication that disruptions of *fbiA-C* would affect INH sensitivity.

Our aim in this study is to cast a wide net in order to find many regions that alter isoniazid resistance and then look at which of these are relevant from an evolutionary point of view. Given that antibiotic sensitivity depends on bacterial metabolism, it is not surprising to find that a sizable portion of the genome can affect antibiotic resistance, even through indirect mechanisms. Previous studies have found similar results in terms of the number of genes that potentially affect resistance. For instance, in the 2017 study by Xu et al. (referenced in the paper) they find 251 regions associated with resistance. However, they have a far less saturated library which means they have less statistical power and as shown in the manuscript our results are widely in agreement with theirs but offering greater detail.

For the analysis of polymorphisms in the clinical isolates, It appears that the "resistance" phenotype is defined to include "resistance to any drug", instead of specifically INH resistance. The authors comment that their results would be similar if they had specifically focused on INH-resistant strains. If that is true, I would strongly recommend they revise their analysis to use only INH-resistance as a phenotype. Otherwise, they risk confounding polymorphisms at loci associated with resistance to other drugs (since there is a lot of co-resistance in these databases). This could lead to false positives in their analysis (i.e. genes not specifically related to INH resistance).

There are a series of reasons why we think including "any-drug" is a better approximation than focussing on INH-only.

- 1) We don't have access to the phenotypes. There may be a non-trivial number of cases where an INH-resistant strain is predicted to be INH-susceptible merely because it does not harbor a known resistance mutation. Furthermore, there may be low-level resistance mutations that do not give a resistant phenotype but may give increased resistance.
- 2) Also, while isoniazid is usually the first resistance to arise, using other antibiotics to annotate resistant branches of the phylogeny can increase our ability to detect infrequent mutations in strains that actually end up acquiring isoniazid resistance.
- 3) Additionally, 78% percent of the mutations classified as occurring on "resistant" branches get the same classification if we consider only isoniazid resistance, which means our result is driven by isoniazid resistance anyways. In accordance with this, results obtained with both approaches strongly correlate (Spearman's $\rho = 0.803$, $p\text{-value} < 2.2e-16$) and the results are very similar but not equal. We believe that results using resistance to any antibiotic and not just isoniazid are more accurate and can help us to better identify less well known isoniazid resistance mechanisms.

Also, why do the authors choose to use a binary definition of resistance (only R and S)? It might have been more informative to take the MIC into account quantitatively (MIC is available some but not all the strains in these databases, and they could have used a subset of strains). This is especially relevant for INH, because it is well-known that some mutations are associated with high-level resistance, and others with low-level resistance.

This could have been interesting but unfortunately there is limited phenotypic information for these strains and even more limited MIC data for these strains, so we would lack any statistical power if we used such an approach.

Specific comments:

Fig 1b: Why is there such a big difference for growth curves for INH treatment between the 2 experiments (libraries)? Can the authors include a scatter plot (as a Supplemental Figure) showing the correlation between mean insertion counts in the genes between the two replicates? (i.e. to assess reproducibility)

The difference is due to a small difference in isoniazid concentration between the two experiments. We have plotted the counts as requested by the reviewer (see below) and determined that both experiments correlate fairly well (Spearman's $\rho = 0.922$, $p\text{-value} < 2.2e-16$).

line 221: Why was the binomial test used instead of the hypergeometric test (Fisher's Exact Test), which is more commonly used for this purpose in the literature on pathway analysis? It could make a difference on which pathways are significant, especially with pathways where only a few genes are involved.

We thank the reviewer for this suggestion. We have revised how each test is applied and concluded that Fisher's exact test works slightly better in this case. We have repeated the analysis and made the necessary changes to the manuscript and supplementary material. The p-values have changed slightly but the conclusions remain unchanged.

Fig 4: How many strains out of ~4,762 actually have *unexplained* INH resistance?

The dataset used for ROC analysis is different from the one used in our phylogenetic association analysis. This one was built from a selection of strains from the CRyPTIC consortium and consisted of 362 strains with known mutations and 82 with no known mutation for isoniazid resistance.

Reviewer #3 (Remarks to the Author):

In this manuscript, the authors use transposon sequencing to comprehensively assess genes associated with susceptibility and resistance of *Mycobacterium tuberculosis* to the first line drug isoniazid. To lend support for a role of the identified genes in isoniazid resistance the authors query a panel of genome sequences of characterized resistant isolates for variations in the respective genes. From the findings presented, the authors infer a new set of genes that are associated with resistance and susceptibility to isoniazid.

Major points:

While this paper presents some potentially important findings relating to novel molecular mechanisms for isoniazid resistance, additional rigor is essential to confirm associations between genotype and phenotype and to assess mechanisms at play. The work could be improved with a more thorough analysis of the data, and with follow-up experiments to confirm association of a few of the top candidate genes in the analysis with isoniazid resistance.

We have incorporated an additional experiment in which we test individual BCG insertion mutants and show that they have altered MICs due to the specific insertion. However, we are unsure about what the reviewer means by more thorough analysis. We have combined functional data with a large dataset of clinical genomes. On one hand, our TNseq approach has produced highly saturated libraries compared to published attempts (see Xu et al. 2017 and DeJesus et al. 2017, both referenced in the paper). And we have analyzed insertions beyond the traditional, dichotomic, straightforward approach essential/non-essential to have a quantitative overview. On the other hand, we have developed a phylogenetic approach which includes mapping of ancestral characters and co-occurrence of mutations in the phylogeny in a robust statistical framework.

Further, use of a single library for the analysis is not viewed as rigorous. It is difficult to follow the details of the Tnseq analysis.

We respectfully disagree with the reviewer that the use of a single library is not seen as rigorous. In this type of studies, the relevant part is the selection experiment and not the creation of the library itself. For instance, Xu et al (2017) use one single library in a very similar study. Furthermore, we checked that our library was similar to other libraries in the same background and the results were in line with other results in the field. For instance, at the start of the Results section we show that our library compares favorably with a set of 14 libraries generated by DeJesus et al (2017).

Display figures are very challenging to understand, in part, due to the extremely small font size that was used, but also due to the limited description of the results in the text (especially Figure 3 and 4).

We have made improvements to font size in some figures of the manuscript in hopes they are better understood now.

Minor points:

line 36, change "curation" to "cure"

line 43 change "resistant" to "resistance"

line 61 change "helper" to "companion"
line 69 change "resistance" to "resistant"
line 128 change "inserted" to "insertion"
line 133 change "it only allows" to "they only allow"
line 135 change "be inserted" to "tolerate insertion"
line 148 change "frequencies" to "abundance"
lines 150-153, awkward wording, perhaps try "Isoniazid-containing cultures show strong enrichment of a fraction of insertions indicating a selective advantage relative to the bulk of the population."
line 156 change "inserted" to "insertion"
line 157 change "independently" to "independent"
line 304 change "be inserted" to "tolerate insertion"
line 416 change "inserted" to "mutated"
line 428 change "inserting" to "insertions in"
line 458 change "inserted" to "insertionally inactivated"

We have made these corrections in the new version of the manuscript.

REVIEWERS' COMMENTS:

Reviewer #1 (Remarks to the Author):

The revised version of the manuscript by Furio et al has been greatly improved. The inclusion of functional data using BCG KO mutants adds to the study and supports their original findings in relation to targets associated with INH resistance.

Overall, the study is interesting and presents an innovative workflow that could be used to study drug resistance to other antibiotics used to treat TB.

Reviewer #2 (Remarks to the Author):

All my concerns have been adequately addressed.

Reviewer #3 (Remarks to the Author):

The authors have adequately addressed the previous comments.

In supplemental Figure 4, it would be useful to the TB community if the authors indicate which genes they have validated to be associated with INH resistance in *M. bovis* BCG.

REVIEWERS' COMMENTS:

Reviewer #1 (Remarks to the Author):

The revised version of the manuscript by Furio et al has been greatly improved. The inclusion of functional data using BCG KO mutants adds to the study and supports their original findings in relation to targets associated with INH resistance. Overall, the study is interesting and presents an innovative workflow that could be used to study drug resistance to other antibiotics used to treat TB.

We thank reviewer 1 for their input and improving the quality of this manuscript.

Reviewer #2 (Remarks to the Author):

All my concerns have been adequately addressed.

We thank reviewer 2 for their input and improving the quality of this manuscript.

Reviewer #3 (Remarks to the Author):

The authors have adequately addressed the previous comments.

In supplemental Figure 4, it would be useful to the TB community if the authors indicate which genes they have validated to be associated with INH resistance in *M. bovis* BCG.

We thank reviewer 3 for their input and improving the quality of this manuscript. Supplementary Figure 4 now contains labels for the most prominent candidate genes tested by using BCG insertion mutants.